# Flavonoids, Phenolic Acids, and Tannin Quantities and Their Antioxidant Activity in Fermented Fireweed Leaves Grown in Different Systems

**DOI:** 10.3390/plants13141922

**Published:** 2024-07-12

**Authors:** Marius Lasinskas, Elvyra Jariene, Jurgita Kulaitiene, Nijole Vaitkeviciene, Ewelina Hallmann, Valdas Paulauskas

**Affiliations:** 1Department of Plant Biology and Food Sciences, Agriculture Academy, Vytautas Magnus University, Donelaicio St. 58, 44248 Kaunas, Lithuania; elvyra.jariene@vdu.lt (E.J.); jurgita.kulaitiene@vdu.lt (J.K.); nijole.vaitkeviciene@vdu.lt (N.V.); 2Bioeconomy Research Institute, Agriculture Academy, Vytautas Magnus University, Donelaicio St. 58, 44248 Kaunas, Lithuania; ewelina_hallmann@sggw.edu.pl; 3Department of Functional and Organic Food, Institute of Human Nutrition Sciences, Warsaw University of Life Sciences, Nowoursynowska 15c, 02-776 Warsaw, Poland; 4Department of Environment and Ecology, Faculty of Forest Sciences and Ecology, Agriculture Academy, Vytautas Magnus University, Donelaicio St. 58, 44248 Kaunas, Lithuania; valdas.paulauskas@vdu.lt

**Keywords:** fermentation, biodynamic, willowherb, antioxidant activity

## Abstract

The increasing demand for organic and biodynamically cultivated fireweeds worldwide has led to a paucity of studies on the effects of solid-phase fermentation and various growth techniques on the quantities of biologically active substances and their antioxidant activity. This experiment was carried out in 2023 at the organic farm in the Jonava district (Safarkos village, Lithuania). The aim of this work was to investigate polyphenols (flavonoids and phenolic acids) and antioxidant activity in fireweed (*Chamerion angustifolium* (L.) Holub) leaves fermented for 24 and 48 h in solid-phase fermentation and natural, organic, and biodynamic cultivation. Fireweeds have high quantities of polyphenols and strong antioxidant activity. The method employed for determining antioxidant activity was spectrophotometric, for measuring polyphenols, high-performance liquid chromatography (HPLC). Principal component analysis (PCA) was used to determine the relationships between the average content of total polyphenols and antioxidant activity in fermented fireweed leaves grown in different systems. This study’s findings demonstrated that the leaves of fireweed cultivated organically had the greatest concentration of total flavonoids, total phenolic acids, and total polyphenols. Comparing the fermentation process effect, the amount of predominant phenolic acids—chlorogenic, p-coumaric, and ellagic acids—as well as the content of oenothein B, during the fermentation process significantly decreased, but the concentration of quercetin-3-O-glucoside after a short time of the fermentation process significantly increased. According to the obtained results, it would be possible to create various health-giving and nature-friendly products from fireweeds.

## 1. Introduction

Farmers, together with scientists and consumers, are searching for ways to cooperate with nature to create various health products from locally grown food and medicinal plants. In addition, they aim to devise cultivation methods that help preserve nature but, at the same time, help provide people with organic food plant raw materials. By applying organic or biodynamically grown farming, as well as the solid-phase fermentation method, it is possible to contribute to a healthier earth and a happier society. Alternative farming systems include biological, biodynamic, organic, and low-cost farms. These farms are lucrative and socially responsible, safeguard the soil resource base, provide enough high-quality food, and are safe for the environment [1]. Numerous studies have shown that organic farming, which avoids the use of synthetic pesticides and fertilizers, can increase soil biological activity and biodiversity when compared to conventional farming [2]. Additionally, it has been observed that compared to traditional agricultural practices, biodynamic farming, a specific kind of organic farming, maintains superior soil quality. The biological characteristics of soil can be influenced by a variety of factors, including plant species [3], soil type [4], and tillage, in addition to different agricultural techniques.

Recently, substances that are physiologically active and exhibit antioxidant activity have drawn a lot of attention from researchers. *Chamerion angustifolium* (L.) Holub (fireweed) is utilized throughout the world. The leaves of fireweed are rich in various secondary metabolites, but polyphenols are the most abundant. The herb of the fireweed contains about 1–2% flavonoids: kaempferol, myricetin, quercetin, and their glucosides. In addition, it contains about 12% tannins: ellagitannin oenothein B and gallic acid derivatives [5]. 

Polyphenols are a class of organic substances characterized by more than one phenolic ring. The number and properties of these phenolic structures influence the various physical, chemical, and biological properties of this class of compounds: flavonoids, phenolic acids, coumarins, stilbenes, and tannins. Polyphenols are mostly of natural origin but can also be synthetic or semi-synthetic. Plant-derived polyphenols have beneficial properties for the human body: anti-inflammatory, anti-allergic, antimicrobial, antiviral, immunomodulating, antioxidant, and others [5]. It is very important to understand the chemical composition and pharmacological effects of fireweed extracts and leaves. However, there is a serious dearth of thorough studies available on this topic. To assess the possible advantages of biodynamically grown cultivation for the quality of fireweed leaves, disparities between organic and biodynamic growth methods must be compared.

When the fireweed plant reaches its maximum flowering stage, at the start of July, when it is synthesizing its most physiologically active compounds, the leaves are typically picked [6]. The high concentration of ellagitannins, particularly oenothein B, has a potent antioxidant impact [7]. Maruska et al. [8] evaluated the flavonoid content of fireweeds gathered throughout the growing season concerning their radical-scavenging action. 

While many people drink black or green tea worldwide, interest in alternative fermented teas prepared from fireweeds is growing. Solid-phase fermentation technology is one of the ways to improve the composition of fireweed leaves to make functional fireweed tea. In addition to biochemical events occurring in the cells during fermentation, there is also a high level of microbial and enzyme activity [9].

The optimization of physiologically active plant compounds’ bioavailability is of critical importance. This fermentation is one of the modern methods for adjusting physiologically active substances and their bioavailability.

It has not yet been studied enough how to apply the biodynamically grown system or how effective it is in comparison to organic systems. Furthermore, no research has been done to examine how various agricultural practices affect the increase in substances in fireweed leaves. To enhance amounts of bioactive molecules, it is imperative to combine the finest agronomic practices.

Important goals for food and nutrition security at the high-level event FOOD 2030: Research & Innovation for Tomorrow’s Nutrition & Food Systems (12–13 October 2016, Brussels) was CLIMATE or the usage of environmentally sustainable and intelligent food systems. In this situation, it is important to use natural resources—soil, water, land, and sea—sustainably within the limits of the planet to preserve them for future generations. In this regard, comparing various farming systems and assessing the advantages of biodynamic cultivation are still important. A strong conviction in the possibility of producing natural and healthy products, particularly those produced in a biodynamic system, stems from the contemporary customer who favors natural farming practices. A distinctive organic farming method known as “biodynamic agriculture” helps address environmental and public health concerns in addition to advancing sustainable farming. Farmers have not yet thoroughly researched and implemented this system’s use or compared its effectiveness to other systems [10].

The aim of this project was to examine the influences of different fermentation technological parameters (different durations: 24 h, 48 h) and natural, organic, and biodynamic cultivation on variation in polyphenols and their antioxidant activity. 

The findings of our study may enable manufacturers of healthy and nutritious foods to create high-quality goods, such as dietary supplements made from the leaf extracts of fireweeds, and farmers may select a more profitable method of cultivating fireweeds based on the ideas of biodynamic farming.

## 2. Results

### 2.1. The Amounts of Biologically Active Compounds

The data obtained indicate that the highest concentration of total polyphenols was found in organic fireweed. The results that were obtained showed statistical significance. The fireweed samples from the biodynamic and natural conditions were identical. The amount of total polyphenols in fireweed leaves decreased depending on the degree of fermentation that was applied. 

All polyphenols were found in significant amounts in the control plants. Following the 24 h fermentation period, a 16.3% drop was seen in the content of every polyphenol. Even after a longer fermentation period, the concentration of total polyphenols was still 59.13% lower than it was after 24 h. In organic samples, significant quantities of all phenolic acids and flavonoids were found. The combination of the biodynamic and natural experiments did not show statistical significance. After 48 h of fermentation, the lowest concentration of both classes of beneficial chemicals was found (Table 1).

In the instance of interaction, we found that the concentration of total polyphenols in fireweed leaves was adversely impacted by the fermentation process. We saw a decrease in total polyphenols in all trial combinations. The organic samples showed the greatest decrease. In comparison to control plants, it was 12.79% after 24 h and 167.4% after 48 h. It was shown that the combination of biodynamic samples experienced the least drop. Compared to control plants, it was 12.41% after 24 h and 19.5% after 48 h. 

Regarding alterations in the phenolic acid concentration, a comparable correlation was noted. The organic samples showed the biggest decline. In comparison to control samples, biodynamic samples lost the least quantity of phenolic acids—only 15.5%—after 48 h. Notably, we saw a significantly larger decrease in total flavonoids after 24 h as opposed to 48 h. However, this was limited to organic and biodynamic samples. When combined naturally, we saw a slight rise in total flavonoids (+11.60%) after 24 h but a decrease in total flavonoid concentration of roughly 9.02% after 48 h (Table 2).

Each phenolic acid was assessed separately. The only organic fireweed leaves that had the noticeably greater gallic acid concentration were those that were organic. The amount of gallic acid was eventually reduced by the fermentation process after 48 h, but it is important to note that during the first 24 h, the significance of the gallic acid concentration rose when the findings were analyzed. 

The greatest and most notable concentration of chlorogenic acid was found in natural samples. Chlorogenic and p-coumaric acids were reduced by prior fermentation, but there were no variations between the 24 and 48 h fermentation times. Organic fireweed leaves had the highest concentration of ellagic and p-coumaric acids that we could find. The content of both acids was not affected by fermentation for 24 h. However, it was considerably reduced over the next 24. 

The biodynamic samples exhibited the highest levels of benzoic acid. After 48 h, we saw an increase in this molecule during the fermentation process, reaching a value that was larger than in the control combination (Table 3). 

We found that in the interaction scenario, the fermentation process only had a beneficial impact on the content of gallic acid. The concentration of gallic acid was noticeably greater after 48 h. This condition was only noted in samples that were biodynamic. Gallic acid levels in organic materials dropped during the 24 h fermentation period, and ultimately, the process increased them in comparison to the initial level. 

Both organic and biodynamic samples reacted similarly to chlorogenic and p-coumaric acids. We saw a decrease in chlorogenic acid after 24 h, although lengthier fermentation produced more fruitful outcomes. Only natural samples displayed negative reactions to the fermentation process after 24 and 48 h.

For ellagic acid, we saw a drop in its concentration in both 24 and 48 h duration. The organic samples showed the greatest drop in content. Benzoic acid’s reaction was a fascinating one. After extended fermentation, a rise in benzoic acid was seen in both natural and biodynamic samples (Table 4).

Seven flavonoids were discovered and measured in all samples (Table 5 and Table 6). The analyzed samples from organic manufacturing included a much higher amount of oenothein B. There were notable distinctions between the samples that were biodynamic and natural. The amount of oenothein B in fireweed leaves decreased as a result of fermentation. In comparison to control samples in leaves, samples with the least amount of this chemical were obtained after 48 h of fermentation (Table 5 and Table 6). 

With regard to quercetin-3-O-rutinoside, it was discovered that the compound was most concentrated in the organic fireweed leaves. Similar to oenothein B which was previously described, there were no variations in the quercetin-3-O-rutinoside level between the natural and biodynamic samples. It appears that fermenting fireweed leaves for a brief period of time is necessary to produce a product with a high concentration of quercetin-3-O-rutinoside. The concentration of rutin increased considerably after 24 h, and its content significantly dropped after another 24 h. 

Myricetin concentrations in organic and natural samples were found to be greater than in natural or biodynamic samples. As we previously mentioned, the goal is to achieve a high myricetin content product by utilizing a brief fermentation period. 

Both organic and natural samples of luteolin had noticeably higher concentrations of this substance. The results show that lengthy fermentation is the most effective way to produce a product with a high luteolin content (Table 5 and Table 6).

Organic samples and organically produced samples were characterized by a higher quantity of quercetin compared to the remainder of the experimental combinations. Using samples of fermented fireweed leaves kept for 48 h was the best approach. 

The largest and statistically significant quantities of quercetin-3-O-glucoside were found in organic materials. It was shown that leaves significantly increased the amounts of quercetin-3-O-glucioside throughout both short and long fermentation times. 

The highest concentration of kaempferol was found in natural samples in fireweed leaves. The amount of kaempferol in fireweed leaves was greatly reduced throughout the fermentation process. The concentration of these flavonoids over the following 24 h was comparable to that following extended fermentation. 

Regarding the natural and organic samples, the initial 24 h fermentation period resulted in a higher concentration of quercetin-3-O-rutinoside in the examined samples. No observations of this phenomenon were made with biodynamic samples. 

In comparison to the long-fermented biodynamic and natural samples, the organic samples had a notably higher amount of rutin after 48 h. It is noteworthy that, of all the studied samples, the greatest decrease in rutin level was seen in the organic samples after 48 h. 

With myricetin, a similar condition to previously reported ones was noted. The concentration of luteolin was not the focus of the fermentation process. The concentration of these chemicals was reduced in all experimental samples by both short (24 h) and long (48 h) fermentation, and only in the biodynamic trial combination was luteolin observed to increase gradually for all fermentation durations in fireweed leaves.

After 48 h, the organic samples had a significantly higher level of rutin than the long-fermented biodynamic and natural samples. Notably, the organic samples showed the biggest drop in rutin levels after 48 h out of all the samples that were examined.

A similar condition to earlier reported ones was seen with myricetin. The fermentation process was not focused on luteolin concentration. Both short (24 h) and long (48 h) fermentation reduced the quantity of these compounds in all experimental samples; luteolin was only shown to gradually increase to all fermentation durations in fireweed leaves in the biodynamic trial combination.

### 2.2. The Antioxidant Activity of Biologically Active Compounds

The results indicated that the samples that were biodynamic and organic exhibited the greatest and most significant levels of antioxidant activity. The characteristics of fireweed leaves’ antioxidant activity were adversely impacted by the fermenting process. Samples without fermentation had the maximum activity, while samples fermented for 48 h displayed the lowest activity (Figure 1).

Upon analyzing the experimental data, we found that the brief 24 h fermentation procedure slowed down the decreasing process. However, among all experimental combinations, natural samples of fireweed leaves showed the least amount of decline. Following more hours of fermentation, organic and biodynamic samples showed a continued decline in antioxidant activity, while natural samples showed the least amount of degradation (Figure 2). 

### 2.3. The Principal Component Analysis

Principal component analysis (PCA) was used to determine the relationships between the average content of total polyphenols and antioxidant activity in fermented fireweed leaves grown in different systems (Figure 3). PCA showed that 100% of the total variance was explained by the first two principal components (PC1 and PC2): PC1 explained 80.06% of the total variability, whereas PC2 explained 19.94%. 

A positive correlation was observed between total phenolics and antioxidant activity. The PCA indicates that total polyphenols were associated with ORG-control and ORG-24 h, which had a positive score along with F1 and F2. Antioxidant activity was associated with NAT-control and BD-control, which had a positive score along with F1 and a negative along with F2 (Figure 3).

## 3. Discussion

Certain researchers claim that any major stressor in the agricultural environment has an impact on phenolic compound accumulation and plant metabolism. If there is a restricted amount of plant nutrients (particularly nitrogen) available during cultivation, the buildup of secondary metabolites in plant tissues may also rise. Stress factors can therefore be produced or avoided by adopting a variety of cultivation and fertilization approaches [11,12]. 

Vaitkeviciene et al. [13] found that tubers cultivated biodynamically had higher amounts of total phenolics and total phenolic acids than tubers grown organically and conventionally. These researchers also found that the total carotenoids in the potato tubers varied depending on the manufacturing system. Compared to organic potatoes, biodynamic potatoes have higher concentrations of this chemical. The epoxidation and de-epoxidation of carotenoids in the xanthophyll cycle may also be influenced by the amount of organic matter present in the soil [14].

The macromolecular compounds found in fireweed leaves are broken down into lower molecular weight substances and secondary metabolite products by the process of solid-phase fermentation and enzymes produced during the metabolism of microorganisms (lactic acid bacteria and yeast), such as polyphenol oxidase, etc. [15]. This process can reduce the quantities of certain compounds. 

However, when solid-phase fermentation is carried out, crushing and pressing the leaves can accelerate the processes that break down the cell walls. This can improve the diffusion of biologically active substances from the inside of the cells, resulting in a more effective compound extraction process. This is believed to have played a significant role in the higher concentrations of polyphenols seen after the solid-phase fermentation process. The data suggest that specific solid-phase fermentation settings may initiate the process of bioactive component accumulation in fireweed leaves [16,17]. 

Fireweed leaves cultivated organically and biodynamically have the highest level of antioxidant activity. Varying manufacturing processes have various effects on the antioxidant activity of the raw materials made from plants, according to data from scientific literature. Higher antioxidant activity was found in Batavia lettuces cultivated using biodynamic production techniques in the study by Heimler et al. [18]. However, when Italian scientists assessed the antioxidant activity of grape berries grown in conventional, biodynamic, and organic systems—Albana and Lambrusco—they discovered no appreciable variations [19].

## 4. Materials and Methods

### 4.1. Chemicals and Reagents

The following were used: ABTS+• (2,2′-azino-bis (3-ethylbenzothiazoline-6-sulfonic acid) (Merck, Warsaw, Poland), acetonitrile (100%) with HPLC purity (Sigma-Aldrich, Pozna, Poland), methanol (100%) with HPLC purity (Sigma-Aldrich, Poland), ultrapure water, polyphenol standards: oenothein B, quercetin-3-O-rutinoside, myricetin, luteolin, quercetin, quercetin-3-O-glucoside, kaempferol, gallic acid, chlorogenic acid, p-coumaric acid, ellagic acid, benzoic acid. All standards were with 99.9% purity, purchased from Sigma-Aldrich, Poland, potassium persulfate from Merck, Poland, and phosphate-buffered saline (PBS solution) from Sigma-Aldrich, Poland.

### 4.2. Field Experiment

#### 4.2.1. Plants Material

The field experiment took place in 2023 at the Giedres Nacevicienes organic farm (No. SER-T-19-00910, Lithuania, 55°00′22 N 24°12′22 E) in the Safarkos hamlet of Jonava district. This agricultural approach has been used to maintain organic fields for the previous ten years; in the last year, biodynamic farming replaced organic farming. In organic farms, the perennial fireweeds were produced for a fourth year. This area of the field was allowed to spontaneously develop fireweed. The experimental plot covered a total of 20 ares. 

In the field trial, fireweed plants were cultivated using organic and biodynamic farming techniques, and their growth was contrasted with that of naturally occurring plants (control). The space between a row of plants was 30 cm, while the space between rows was 70 cm. A compost of 25 t ha^−1^ for organic farming or 25 t ha^−1^ for biodynamic farming was applied to the soil in the inter-row areas two weeks prior to the onset of plant growth (mid-June).

Biodynamic preparation 500 (BD) was sprayed on biodynamic experimental fields in May of the second decade at a concentration of 1% solution (200 L ha^−1^). During the vegetation period, BD preparation 501 0.5% solution was sprayed on fireweed leaves twice: once in the morning during the stage of leaf formation (middle of June) and once at the start of plant mass flowering (July, first decade) (200 L ha^−1^). 

Water was sprayed on organic and natural experimental fields in May of the second decade. During the vegetation period, water was sprayed twice on fireweed leaves: once in the morning at the onset of leaf formation (mid-June) and once in the morning at the start of plant mass flowering (July, first decade). 

There was no use of plant protection agents against diseases or pests. Using the same techniques as European biodynamic farms, BD preparations were sprayed on the fireweed plants and soil in the biodynamic system (Demeter International, 2013). The experiment employed biodynamic compost and BD preparations 500 and 501, which were acquired from an Internationale Biodynamische Praparatezentrale (CvW KG, Darmstadt, Germany), a Demeter-certified farm [20].

In this experiment, during 2023, the air temperature was higher than the standard climate norm (except for the month of May), and the precipitation was much less than the standard climate norm (except for the month of March). So, it could be concluded that 2023 was warm and dry. 

The data from our previous research year 2022 have already been published in the journal Plants [21]. In the experiment, during the vegetation of fireweeds in 2022, the air temperature was higher than the standard climate norm (except for the month of May), and the precipitation was more than the standard climate norm (except for the month of March). So, for comparison, we found that during the vegetation of plants in 2022, it was warm and wet. Analyzing the results of two years, we can say that the amount of polyphenols depends on the meteorological conditions during the vegetation of the plant.

#### 4.2.2. Experimental Design

At the start of fireweed flowering (first decade of July), three replications (field plots) of fireweeds were randomly selected for leaf collection in each farming system. In total, 10.8 kg of leaves were sampled for each agricultural system: -Naturally produced (control): 3.6 kg for a 24 h solid-phase fermentation, 1.2 kg for a 48 h fermentation, and 3.6 kg for no fermentation.-Fireweed cultivated organically: 3.6 kg (unfermented) for control, 3.6 kg for a 24 h solid-phase fermentation, and 3.6 kg for a 48 h fermentation.-Fireweed cultivated biodynamically: 3.6 kg (unfermented) for control, 3.6 kg for a 24 h solid-phase fermentation, and 3.6 kg for a 48 h fermentation.-Control (0 h): the leaves were kept for the anticipated amount of time but had not been fermented.

Using specialized plastic blades, fresh fireweed leaves were chopped for the solid-phase fermentation process. The resultant raw material was then split into three 1.200 kg subsamples. The fermentation-ready bulk was firmly packed into glass jars and sealed with an air-passing lid. For 24 and 48 h, the fermentation process was conducted in the chamber at 30 °C.

Following fermentation, the raw materials were lyophilized in a ZIRBUS sublimation dryer 3 × 4 × 5 (ZIRBUS Technology, Bad Grund, Germany) and frozen at−35 °C. For further analysis, the lyophilized leaves were ground into a powder using a Grindomix GM 200 laboratory mill (Retsch GmbH, Haan, Germany) (Figure 4). 

### 4.3. Laboratory Analyses

#### 4.3.1. Polyphenol Identification and Quantification

Weighing out 100 mg of freeze-dried plant material, we added 5 mL of 80% methanol, vortexed (using a Micro-Shaker 326 M from Premeo, Marki, Poland), incubated for 10 min at 30 °C and 5500 Hz in an ultrasonic bath, and centrifuged (10 min at 3780× *g* and 5 °C). Following a second centrifugation (5 min, 31.180× *g*, 0 °C) of the acquired supernatants, 900 µL of the clear supernatants were transferred to HPLC vials for analysis. The HPLC technique was utilized to identify and separate phenolic compounds, which include flavonoids, phenolic acids, and flavanols. The Shimadzu kit, which includes two LC-20AD pumps, a CBM-20A controller, a SIL-20AC column oven, and a UV/Vis SPD-20 AV spectrometer, was manufactured by USA Manufacturing Inc. in the Broadview, IL, USA. On a Synergy Fusion-RP 80i Phenomenex column (250 × 4.60 mm), the phenolic compounds were separated at a flow rate of 1 mL min^−1^. The two gradient phases (acidified with ortho-phosphoric acid, pH 3.0) were 10% (*v*/*v*) acetonitrile and ultrapure water (phase A) and 55% (*v*/*v*) acetonitrile and ultrapure water (phase B). A total of 38 min were spent on the analysis in total, using the phase–time software below: Phase A and B percentages range from 1.00 to 22.99 min, 50% phase A and 5 % phase B, 80% phase A and 20% phase B, and 95% phase A and 5% phase B from 29.00 to 38.00 min. For flavanols, the wavelength was λ = 250 nm, while for phenolic acids, it was λ = 370 nm. Using 99.9% pure standards (Sigma-Aldrich, Poland) and the standards’ designated analytical periods, the phenolic compounds were identified (all chromatograms are in Appendix A).

#### 4.3.2. Antioxidant Activity

The colorimetric spectrophotometric approach using ABTS+• (2,2′-azino-bis (3-ethylbenzothiazoline-6-sulfonic acid) cation radicals was employed to determine the antioxidant activity. The modified protocol by Re et al. (1999) [22] calls for dissolving 0.0384 g of ABTS radical reagent in 5.0 mL of deionized water, followed by the addition of 5.0 mL of aqueous potassium persulfate (K_2_S_2_O_8_), which was made by dissolving 0.026 g in 20 mL of deionized water. The resulting ABTS+• radical solution (10 mL) was then incubated at 21 °C in the dark for 12 h and lastly diluted with phosphate-buffered saline (PBS solution) to obtain a blank sample absorbance of 0.700 ± 0.02 at λ = 734 nm. The tested samples’ water extracts, which were synthesized in accordance with the preceding description (point 2.5.), were measured. 

The antioxidant activity was measured by dispensing 1.5 mL (diluted with PBS solution) into 10 mL glass test tubes, adding 3.0 mL of ABTS+• cation radical solution (with a predetermined absorbance of 0.700 ± 0.02), and incubating for 6 min at 21 °C. The absorbance was then measured at a wavelength of λ = 734 nm in a spectrophotometer (UV–VIS UV-610, Metash Instruments Co., Ltd., Shanghai, China). After taking into consideration the applied dilutions, the determination was carried out nine times independently. The results were then represented as µM TEAC 100 g^−1^ D.W. using the calibration curve (y = −5.6017 × 0.7134, R^2^ = 0.9998) for Trolox as the reference material.

### 4.4. Statistical Analysis

The statistical software Statgraphics Centurion 15.2.11.0 (StatPoint Technologies, Inc., Warrenton, VA, USA) was used to evaluate the findings that were collected. The average values of nine (n = 9) individual measurements are displayed in the tables for the biodynamic, organic, and natural production system as well as the two fermentation durations (24 and 48 h) in relation to the non-fermented control. After doing a two-way analysis of variance using Tukey’s test, differences between the groups were deemed statistically significant at the *p*-value of less than 0.05. Additionally included for every mean value in the tables is the standard error (SE). Principal component analysis (PCA) was used to determine the relationships between the total polyphenols and antioxidant activity in fermented fireweed leaves grown in different systems utilizing the software XLSTAT (XLSTAT, 2018; New York, NY, USA).

## 5. Conclusions

The results that were reviewed indicated that the levels of biologically active and antioxidant activity in fireweed leaves were influenced by solid-phase fermentation and various growth regimes.

The findings showed that non-fermented (control) and fermented 24 h leaves of fireweed cultivated organically had the greatest concentration of total flavonoids, total phenolic acids, and total polyphenols. 

Comparing the fermentation process effect, the amount of predominated phenolic acids, chlorogenic, p-coumaric, and ellagic acids, as well as the content of oenothein B, during the fermentation process significantly decreased, but the concentration of quercetin-3-O-glucoside after a short time of the fermentation process significantly increased.

The results indicate that the samples that were biodynamic and organic exhibited the greatest and most significant levels of antioxidant activity. The characteristics of fireweed leaves’ antioxidant activity were adversely impacted by the fermenting process.

It can be suggested that organically grown, non-fermented fireweed leaves may be a promising raw material for the development of functional foods or medicinal preparations.

## Figures and Tables

**Figure 1 plants-13-01922-f001:**
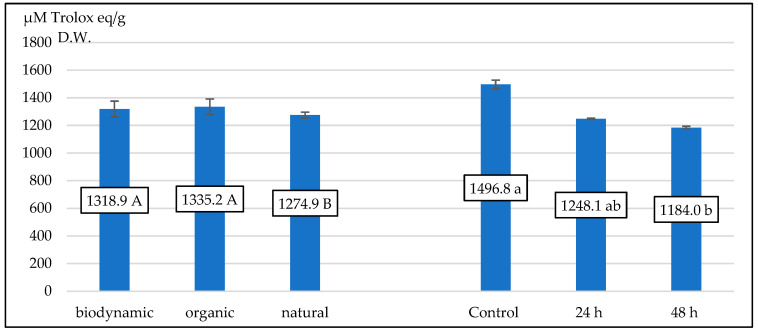
The mean value for antioxidant activity (in µM Trolox eq./g D.W.) for different fireweed leaves. Means on bars followed by the same letter are not significantly different at the 5% level of probability (*p* < 0.05).

**Figure 2 plants-13-01922-f002:**
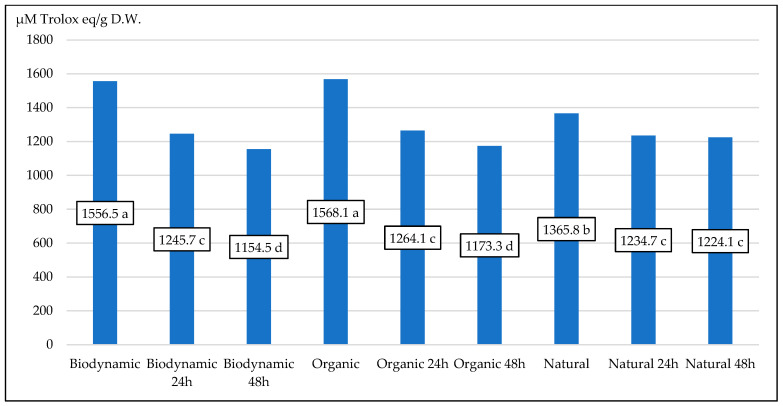
The antioxidant activity (in µM Trolox eq./g D.W.) for different fireweed leaves according to the fermentation process. Means on bars followed by the same letter are not significantly different at the 5% level of probability (*p* < 0.05).

**Figure 3 plants-13-01922-f003:**
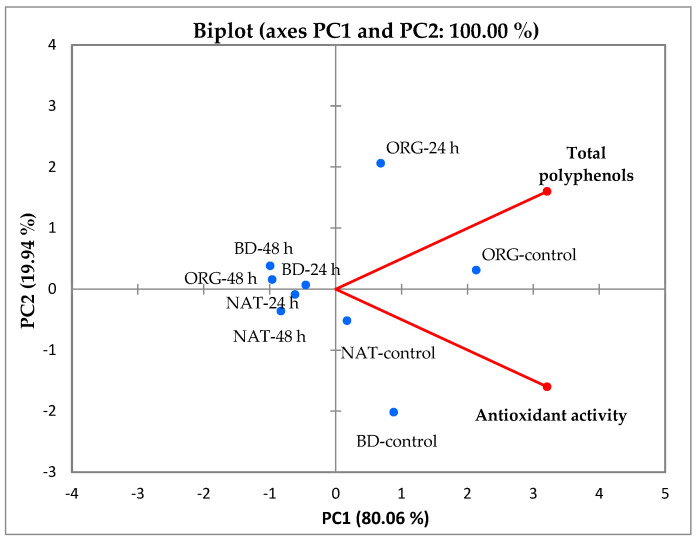
Principal component analysis of total polyphenols and antioxidant activity in fermented fireweed leaves (control—unfermented leaves, 24 h—fermentation 24 h, 48 h—fermentation 48 h) grown in different systems (BD—biodynamic, ORG—organic, NAT—natural).

**Figure 4 plants-13-01922-f004:**
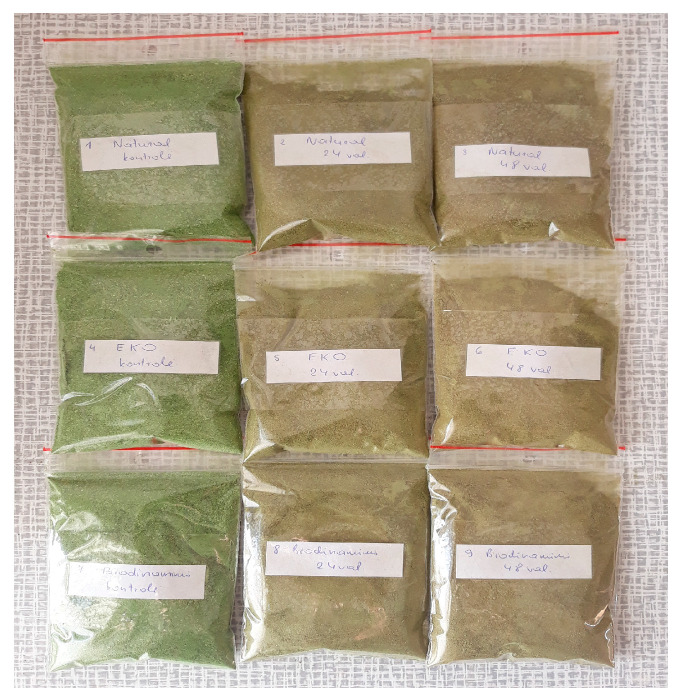
Fermented and lyophilized fireweed leaves (powders).

**Table 1 plants-13-01922-t001:** The mean value for the sum of identified polyphenols, phenolic acids, and flavonoids (in mg/100 g D.M.) in fireweed according to different production systems and duration times.

Mean Values	Sum of Identified Polyphenols, mg/100 g	Sum of Identified Phenolic Acids, mg/100 g	Sum of Identified Flavonoids, mg/100 g
biodynamic	2465.88 ± 101.17 ^1^ B ^2^	1665.14 ± 52.21 B	800.74 ± 53.11 B
organic	4042.35 ± 489.77 A	3003.60 ± 428.77 A	1038.75 ± 68.77 A
natural	2321.69 ± 135.00 B	1562.52 ± 139.39 B	759.17 ± 14.87 B
Control	3674.99 ± 399.50 a	2684.37 ± 333.98 a	990.62 ± 77.88 a
24 h	3165.61 ± 379.18 a	2267.21 ± 345.23 a	898.40 ± 35.84 b
48 h	1989.33 ± 36.52 b	1279.68 ± 57.33 b	709.65 ± 30.30 c
*p*-value			
production system	<0.0001	<0.0001	<0.0001
duration	<0.0001	<0.0001	<0.0001

^1^ Data are presented as the mean ± SE (standard error) with ANOVA *p*-value. ^2^ Means in column followed by the same letter are not significantly different at the 5% level of probability (*p* < 0.05).

**Table 2 plants-13-01922-t002:** The content of the sum of identified polyphenols, phenolic acids, and flavonoids (in mg/100 g D.M.) in fireweed according to different production systems and duration times.

Production System	Duration, Hours	Sum of Identified Polyphenols, mg/100 g	Sum of Identified Phenolic Acids, mg/100 g	Sum of Identified Flavonoids, mg/100 g
biodynamic	Control	2807.64 ± 50.81 ^1^ b ^2^	1836.87 ± 50.86 bc	970.77 ± 0.31 b
	24 h	2498.73 ± 8.46 c	1654.95 ± 8.46 c	843.78 ± 0.08 c
	48 h	2091.28 ± 58.28 d	1503.60± 58.28 c	587.68 ± 1.16 e
organic	Control	5365.18 ± 70.17 a	4089.38 ± 70.17 a	1275.80 ± 57.66 a
	24 h	4755.79 ± 136.34 a	3714.36± 136.34 a	1041.42 ± 32.99 b
	48 h	2006.10 ± 17.91 d	1207.06 ± 17.91 e	799.04 ± 15.53 d
natural	Control	2852.14 ± 14.79 b	2126.85 ± 14.79 b	725.30 ± 23.49 d
	24 h	2242.32 ± 7.32 c	1432.33 ± 7.32 d	809.99± 7.23 bc
	48 h	1870.61 ± 5.67 e	1128.38 ± 5.67 e	742.22 ± 6.81 d
*p*-Value				
production × duration		<0.0001	<0.0001	<0.0001

^1^ Data are presented as the mean ± SE (standard error) with ANOVA *p*-value. ^2^ Means in column followed by the same letter are not significantly different at the 5% level of probability (*p* < 0.05).

**Table 3 plants-13-01922-t003:** The mean value for individual identified phenolic acids (in mg/100 g D.M.) in fireweed according to different production systems and duration times.

Mean Value	Gallic, mg/100 g	Chlorogenic, mg/100 g	*P*-Coumaric, mg/100 g	Ellagic, mg/100 g	Benzoic, mg/100 g
biodynamic	21.36 ± 2.13 ^1^ B ^2^	25.66 ± 1.35 B	50.00 ± 1.93 B	1549.34 ± 57.78 B	18.78 ± 1.70 A
organic	32.66 ± 1.13 A	28.55 ± 0.71 B	142.14 ± 3.93 A	2788.41 ± 432.19 A	11.84 ± 192 B
natural	24.34 ± 1.58 B	40.26 ± 1.87 A	38.59 ± 2.61 C	1453.88 ± 137.36 B	5.46 ± 0.53 C
Control	24.72 ± 3.18 b	35.29 ± 2.53 a	72.15 ± 13.48 b	2540.80 ± 318.38 a	11.41 ± 1.74 a
24 h	29.35 ± 1.25 a	29.62 ± 3.03 b	77.70 ± 15.11 b	2121.15 ± 331.66 a	9.39 ± 2.56 b
48 h	24.30 ± 1.58 b	29.55 ± 1.07 b	80.88 ± 18.12 a	1129.68 ± 61.42 b	15.28 ± 2.27 a
*p*-value					
production system	<0.0001	<0.0001	<0.0001	<0.0001	<0.0001
duration	<0.0001	<0.0001	<0.0001	<0.0001	<0.0001

^1^ Data are presented as the mean ± SE (standard error) with ANOVA *p*-value. ^2^ Means in a column followed by the same letter are not significantly different at the 5% level of probability (*p* < 0.05).

**Table 4 plants-13-01922-t004:** The content of individual identified phenolic acids (in mg/100 g D.M.) in fireweed according to different production systems and duration times.

Production System	Duration, Hours	Gallic, mg/100 g	Chlorogenic, mg/100 g	*P*-Coumaric, mg/100 g	Ellagic, mg/100 g	Benzoic, mg/100 g
biodynamic	Control	12.55 ± 0.05 ^1^ d ^2^	30.82 ± 0.09 b	53.54 ± 1.00 c	1728.01 ± 51.28 c	11.95 ± 0.07 c
	24 h	25.42 ± 1.36 b	20.96 ± 0.07 e	41.96 ± 0.32 d	1546.37 ± 8.94 d	20.24 ± 0.13 a
	48 h	26.12 ± 0.32 b	25.19 ± 0.24 d	54.50 ± 0.17 c	1373.65 ± 57.98 e	24.15 ± 0.21 a
organic	Control	35.36 ± 1.78 a	29.07 ± 0.22 c	128.24 ± 0.96 b	3879.25 ± 11.65 a	17.47 ± 0.56 b
	24 h	33.75 ± 0.80 a	25.77 ± 0.38 d	141.64 ± 0.06 a	3509.23 ± 130.05 a	3.98 ± 0.04 f
	48 h	28.89 ± 0.17 ab	30.80 ± 0.13 bc	156.53 ± 2.14 a	976.76 ± 13.70 f	14.08 ± 0.26 bc
natural	Control	26.24 ± 0.25 b	45.97 ± 0.26 a	34.67 ± 0.03 e	2015.15 ± 8.64 b	4.82 ± 0.01 e
	24 h	28.88 ± 0.02 ab	42.14 ± 0.32 a	49.50 ± 0.09 c	1307.85 ± 12.03 e	3.97 ± 0.24 f
	48 h	17.89 ± 0.70 c	32.67 ± 0.26 b	31.60 ± 0.59 e	1038.63 ± 2.23 f	7.59 ± 0.23 d
*p*-value						
production × duration		<0.0001	<0.0001	<0.0001	<0.0001	<0.0001

^1^ Data are presented as the mean ± SE (standard error) with ANOVA *p*-value. ^2^ Means in a column followed by the same letter are not significantly different at the 5% level of probability (*p* < 0.05).

**Table 5 plants-13-01922-t005:** The mean value for individually identified flavonoids (in mg/100 g D.M.) in fireweed according to different production systems and duration times.

Mean Values	Oenothein B, mg/100 g	Quercetin-3-O-Rutinoside, mg/100 g	Myricetin, mg/100 g	Luteolin, mg/100 g	Quercetin, mg/100 g	Quercetin-3-O-Glucoside, mg/100 g	Kaempferol, mg/100 g
biodynamic	727.27 ± 55.76 ^1^ B ^2^	14.88 ± 0.30 B	8.86 ± 0.50 B	3.38 ± 0.24 A	1.95 ± 0.07 C	41.34 ± 2.27 B	3.06 ± 0.06 C
organic	869.57 ± 72.93 A	58.90 ± 11.63 A	15.96 ± 0.81 A	2.55 ± 0.08 B	2.28 ± 0.08 B	84.39 ± 6.11 A	5.10 ± 0.32 B
natural	671.66 ± 14.08 C	17.09 ± 0.54 B	14.65 ± 0.78 A	3.55 ± 0.10 A	3.83 ± 0.10 A	41.85 ± 1.43 B	6.54 ± 0.26 A
Control	899.26 ± 71.98 a	24.91 ± 4.34 b	13.05 ± 1.51 b	3.02 ± 0.23 b	2.41 ± 0.23 b	42.84 ± 3.80 b	5.13 ± 0.45 a
24 h	766.47 ± 17.03 b	46.73 ± 14.28 a	14.58 ± 1.21 a	3.06 ± 0.09 b	2.37 ± 0.09 b	60.42 ± 8.38 a	4.78 ± 0.60 a
48 h	602.76 ± 23.83 c	19.23 ± 1.65 c	11.85 ± 0.72 c	3.41 ± 0.26 a	3.27 ± 0.26 a	64.33 ± 8.15 a	4.79 ± 0.53 a
*p*-value							
production system	<0.0001	<0.0001	<0.0001	<0.0001	<0.0001	<0.0001	<0.0001
duration	<0.0001	<0.0001	<0.0001	<0.0001	<0.0001	<0.0001	n.s.

^1^ Data are presented as the mean ± SE (standard error) with ANOVA *p*-value. ^2^ Means in a column followed by the same letter are not significantly different at the 5% level of probability (*p* < 0.05).

**Table 6 plants-13-01922-t006:** The content of individual identified flavonoids (in mg/100 g D.M.) in fireweed according to different production systems and duration times.

Production System	Duration, Hours	Oenothein B, mg/100 g	Quercetin-3-O- Rutinoside, mg/100 g	Myricetin, mg/100 g	Luteolin, mg/100 g	Quercetin, mg/100 g	Quercetin-3-O- Glucoside, mg/100 g	Kaempferol, mg/100 g
biodynamic	Control	907.44 ± 0.35 ^1^ a ^2^	15.68 ± 0.23 d	6.97 ± 0.05 f	2.76 ± 0.01 d	1.86 ± 0.02 d	32.75 ± 0.29 e	3.30 ± 0.03 c
	24 h	769.93 ± 0.58 b	13.68 ± 0.8 e	10.60 ± 0.13 de	3.00 ± 0.02 c	1.76 ± 0.02 e	41.93 ± 0.38 d	2.88 ± 0.02 d
	48 h	504.43 ± 0.97 d	15.28 ± 0.06 d	9.01 ± 0.17 e	4.39 ± 0.01 a	2.22 ± 0.01 c	49.35 ± 0.20 c	2.99 ± 0.02 d
organic	Control	1148.72 ± 56.82 a	43.32 ± 0.46 ab	14.34 ± 0.18 b	2.34 ± 0.06 d	2.03 ± 0.03 c	58.61 ± 0.46 b	6.44 ± 0.07 ab
	24 h	809.96 ± 34.23 b	107.21 ± 0.76 a	19.37 ± 0.13 a	2.81 ± 0.08 d	1.88 ± 0.03 d	95.86 ± 1.46 a	4.33 ± 0.13 b
	48 h	650.02 ± 15.44 c	26.17 ± 0.05 b	14.18 ± 0.21 b	2.49 ± 0.06 d	2.94 ± 0.12 c	98.70 ± 0.26 a	4.55 ± 0.01 b
natural	Control	641.63 ± 22.59 c	15.74 ± 0.07 d	17.82 ± 0.16 ab	3.94 ± 0.03 b	3.35 ± 0.11 b	37.16 ± 1.38 e	5.65 ± 0.17 b
	24 h	719.52 ± 8.26 b	19.28 ± 0.21 c	13.76 ± 0.17 c	3.37 ± 0.07 bc	3.47 ± 0.16 b	43.45 ± 0.82 d	7.13 ± 0.33 a
	48 h	653.83 ± 5.93 c	16.23 ± 0.38 d	12.37 ± 0.18 d	3.35 ± 0.07 bc	4.66 ± 0.17 a	44.95 ± 2.09 d	6.83 ± 0.23 ab
*p*-value								
production × duration		<0.0001	<0.0001	<0.0001	<0.0001	0.0092	<0.0001	<0.0001

^1^ Data are presented as the mean ± SE (standard error) with ANOVA *p*-value. ^2^ Means in a column followed by the same letter are not significantly different at the 5% level of probability (*p* < 0.05).

## Data Availability

Data are contained within the article and Appendix A.

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
