# Peer review of "Flavonoids, Phenolic Acids, and Tannin Quantities and Their Antioxidant Activity in Fermented Fireweed Leaves Grown in Different Systems"

_plants, 2024, doi:10.3390/plants13141922_

Round 1

Reviewer 1 Report

Comments and Suggestions for Authors

The manuscript is dedicated to the study of the influence of fermentation time and cultivation on the production of polyphenols, and their antioxidant activity.

Generally, the paper is a good experimental study that, I believe, will be interesting to readers. However, the manuscript needs significant improvement (see below).

Comments:

1. The abstract must be improved. Please make the introduction part shorter (first two sentences), continue the sentences, and add more details from the obtained results.

2. Add the clear aims of the study. Normally, the end of Introduction is a good place for this.

3. Tables must be improved:

- add units for the presented values,

- inprove the the head of the tables (may be add onem more line),

4. Check the Figures for format of numbers (use English standard).

5. Figures can be builded - make different colored groups according the time or cultivation.

6. Lines 340-347: is thete the list? Or plain text?

7. Please extend the Conclusions with own results.

Comments on the Quality of English Language

Moderate editing of the English language is needed.

Reviewer 2 Report

Comments and Suggestions for Authors

Manuscript Title: Flavonoids, Phenolic acids, and Tannins Quantities and Their 2 Antioxidant Activity in Fermented Fireweed Leaves, Grown in 3 Different Systems.

General comments:

This is an interesting manuscript concerned with the influences of different fermentation technological parameters (i.e. durations of 24 and 48 hours) and natural, organic, and biodynamic cultivation on the variation of polyphenols, and their antioxidant activity from fireweeds. However, it needs modifications before it can be accepted for publication. Below, please, see my suggestions and comments for each section of the manuscript.   

Abstract

1. Write the scientific name of Fireweed.

2. Rewrite this portion and write one line about the importance of firewood.

3. (Page 1; line- 24) Changed the line “The purpose of this study – examine the influences of different fermentation technological parameters (different durations: 24 hours, 48 hours) and natural, organic, and biodynamic cultivation on the variation of polyphenols, and their antioxidant activity” by “The aim of this work was to investigate polyphenols (flavonoids, and phenolic acids) and antioxidant activity in fireweed leaves fermented for 24, and 48 hours in solid-phase fermentation and natural, organic, and biodynamic cultivation”.

4. (Page 1; line- 28) Changed the line “The study's findings demonstrated that this type of fermentation and various cultivation environments, including organic, biodynamic, and naturally occurring ones, had a major effect on the composition of leaves compounds”. Write the result.

Results

1.     In the HPLC analysis, the author should include a chromatogram that displays the retention periods of flavonoids and phenolic compounds in fireweed.

2.     I recommend including the correlation between total polyphenols and antioxidant  activity using principal component analysis. It will enhance the quality of this article.

Material and methods

1.     Write chemicals and reagents used.

2.     (Page 9; line- 303; 4.1. Field Experiment) Field Experiment portion should divided into two parts- 1. Plants material, 2. Experimental design.

3.      The author calculates the amount of flavonoids and phenolic acids using HPLC.  Write down the detailed methods used to quantify total flavonoids, total phenolic acids, and total polyphenols.

Conclusion

1. In the conclusion, the author should specify the duration (24 or 42 hours) during which the highest amount of polyphenolic compounds were extracted.

Comments on the Quality of English Language

Improvement is required

Reviewer 3 Report

Comments and Suggestions for Authors

The submitted manuscript under the title  „Flavonoids, Phenolic acids, and Tannins Quantities and Their Antioxidant Activity in Fermented Fireweed Leaves, Grown in Different Systems authors: Marius Lasinskas, Elvyra Jariene, Jurgita Kulaitiene , Nijole VaitkevicienÄ—  and Ewelina Hallmann is a very interesting for pubication.

I have more questions for the authors, so I ask them for additional clarifications.

I believe that the introduction must be significantly improved. First of all, I think that literature data on the content and amount of polyphenols for the examined plant should be included.

I also think that these investigations should have covered two years and not just one. The amount of polyphenolic compounds changes drastically depending on the climatic conditions of the year the plant is grown.

In line 307, it appears as an area unit of acres. Correct in ares to hectares.

I believe that the authors should provide more details about the applied standards. Were all the compounds for which they had standards identified.

I also wonder on what principle they chose the mentioned standards.

In line 100 onwards they list the amounts of total polyphenols, total phenolic acids and total flavonoids. Is it the sum of the quantities obtained by the HPLC method.

I also believe that in the results and discussion, the content and quantity of polyphenols should be discussed in accordance with the literature data for the examined plant.

Round 2

Reviewer 1 Report

Comments and Suggestions for Authors

The manuscript was improved. However, tables and figures still need some attention.

Tables:

1. Please avoid repetition of the same text and digital values in tables, if possible. Find the suggested design for the tables in the attachment. Apply to all tables.

2. The caption for Fig. 2 is missing.

Figures:

Error bars are still not clearly visible. I suggest changing the color of the lines from gray to black and increasing the line thickness if necessary.

Comments on the Quality of English Language

Reviewer 3 Report

Comments and Suggestions for Authors

The authors responded positively to all objections. I agree to accept the work in the specified form for publication.

Author Response

Thank You very much!